# Efficacy and Safety of Auricular Acupuncture for the Treatment of Insomnia in Breast Cancer Survivors: A Randomized Controlled Trial

**DOI:** 10.3390/cancers13164082

**Published:** 2021-08-13

**Authors:** Melanie D. Höxtermann, Katja Buner, Heidemarie Haller, Wiebke Kohl, Gustav Dobos, Mattea Reinisch, Sherko Kümmel, Holger Cramer, Petra Voiss

**Affiliations:** 1Department of Internal and Integrative Medicine, Evang. Kliniken Essen-Mitte, Faculty of Medicine, University of Duisburg-Essen, 45276 Essen, Germany; m.hoextermann@kem-med.com (M.D.H.); katja.buner@stud.uni-duisburg-essen.de (K.B.); h.haller@kem-med.com (H.H.); w.kohl@kem-med.com (W.K.); gustav.dobos@uni-due.de (G.D.); h.cramer@kem-med.com (H.C.); 2Breast Unit, Evang. Kliniken Essen-Mitte, 45136 Essen, Germany; m.reinisch@kem-med.com (M.R.); s.kuemmel@kem-med.com (S.K.); 3Charité–Universitätsmedizin Berlin, Department of Gynecology with Breast Center, 10117 Berlin, Germany

**Keywords:** breast neoplasms, sleep initiation and maintenance disorders, acupuncture, ear, cancer survivors, randomized controlled trial, clinical study, individualized treatment

## Abstract

**Simple Summary:**

Sleep problems impair up to 70% of breast cancer survivors. We studied whether ear acupuncture improves sleep quality in breast cancer survivors with sleeplessness. Fifty-two female breast cancer survivors were randomly distributed to either 10 group ear acupuncture sessions or a onetime psychoeducation group. We found that sleep quality improved over the 5 weeks of the study in the ear acupuncture group in comparison to the psychoeducation group. Stress, anxiety and fatigue improved as well. These effects were lost after 17 and 29 weeks.

**Abstract:**

Among women, breast cancer is the most commonly diagnosed cancer worldwide. Sleep problems impair 40–70% of breast cancer survivors. This randomized controlled trial evaluates the effect of auricular acupuncture on sleep quality in breast cancer survivors suffering from insomnia. Fifty-two female breast cancer survivors with insomnia (mean age 55.73 ± 8.10 years) were randomized either to 10 treatments of auricular acupuncture within five weeks (*n* = 26), or to a single session of psychoeducation plus an insomnia advice booklet (*n* = 26). The primary outcome was sleep quality (measured by the Pittsburgh Sleep Quality Index) at week 5. Secondary outcomes were inflammation parameter (interleukin-6), stress, anxiety, depression, quality of life, and fatigue at week 5, and sleep quality, stress, anxiety, depression, quality of life, and fatigue 17 and 29 weeks after randomization. Intention-to-treat analysis showed a significantly stronger increase in sleep quality in the auricular acupuncture group compared to the psychoeducation group (*p* = 0.031; *η*^2^_p_ = 0.094) at week 5. Furthermore, auricular acupuncture improved stress (*p* = 0.030; *η*^2^_p_ = 0.094), anxiety (*p* = 0.001; *η*^2^_p_ = 0.192), and fatigue (*p* = 0.006; *η*^2^_p_ = 0.148) at week 5 compared to psychoeducation. No significant group difference was found concerning the other outcomes at week 5, or in any outcome at week 17 or week 29. No serious adverse events occurred during the study period. In conclusion, a semi-standardized group auricular acupuncture might be an effective and safe intervention in treating insomnia in breast cancer survivors in the short term, and may reduce stress, anxiety, and fatigue as well. Long-term effects remain questionable.

## 1. Introduction

Among females, breast cancer is the most commonly diagnosed cancer, with an incidence of 11.6% worldwide in 2018 [1], and a 10-year survival rate exceeding 80% [2]. About 40–70% of breast cancer survivors report sleep problems even years after diagnosis, depending on the applied diagnostic criteria [3,4,5]. A common sleep disorder in breast cancer patients is insomnia [6]. The DSM-5 defines insomnia as a dissatisfaction with sleep quantity or quality, associated with difficulty initiating sleep or maintaining sleep, with the inability to return to sleep, present for at least 3 months, and occurring at least 3 nights per week despite adequate opportunity for sleep [7].

Insomnia leads to impaired psychological and physical health [8,9], and might even increase mortality in breast cancer patients [10,11]. Sleep influences the hypothalamic–pituitary–adrenal axis and the sympathetic nervous system which, in turn, regulate adaptive and innate immune responses [12]. While sleep disturbances seem to induce a downregulation of adaptive immunity, the innate immune responses are upregulated with increases in cellular and genomic markers of inflammation. A loss of total sleep time leads to daytime increases in circulating interleukin 6 (IL-6) levels [12,13]. Furthermore, circadian control seems to influence the gene expression of cytokines, including IL-6 [14].

The treatment of insomnia is typically pharmacological [15,16,17], even though current international guidelines recommend drug treatment only for short-term treatment of insomnia (≤4 weeks) [18,19]. Moreover, side effects as well as drug-to-drug interactions in case of polypharmacy and dependency need to be taken into account [20,21]. Instead, cognitive behavioral therapy for insomnia (CBT-I) is recommended as a first-line treatment for chronic insomnia [19]. Further treatment options for insomnia are warranted, since CBT-I is often not available for patients, and poor adherence can decrease its effectiveness [15,22]. Psychoeducation is a central element of CBT-I [23]. A single psychoeducation session that addresses sleep-related maladaptive thoughts and beliefs, as well as sleep perception, seems to improve subjective sleep perception in patients with insomnia [23].

As a widely used, relatively safe, and well-accepted intervention, acupuncture might be a promising treatment option, which appears to be superior to sham, drugs, or hormone therapy in improving cancer-related insomnia [24,25,26,27]. Auricular acupressure significantly improved sleep in ovarian cancer patients compared to a group receiving sleep hygiene advice [28]. The National Cancer Institute defines auricular acupuncture as “a type of acupuncture in which thin needles are inserted at specific points on the outer ear to control pain and other symptoms.” [29]. It is assumed that the human body is represented homunculus-like on the ear [30,31]. The clinical specificity of auricular points/areas representing organs or structures of the body is not yet well established, but the evidence is growing [30,31]. Stimulation by a needle seems to lead to specific activation in the brain, mainly by the auricular branch of the vagus nerve [31]. Two reviews showed auricular acupuncture to be a relatively safe, time- and cost-efficient intervention that can be offered in a group setting [26,32]. Therefore, auricular acupuncture is easier to implement in clinical settings than body acupuncture. In oncology, auricular therapy has been studied—especially in the treatment of cancer-related pain, but also for constipation—and appears to be effective and safe [33,34,35,36,37].

This randomized trial examines the effect of auricular acupuncture versus psychoeducation on sleep quality in breast cancer survivors with an insomnia diagnosis. Furthermore, the effects on inflammation parameter (IL-6), stress, anxiety, depressive symptoms, quality of life, and fatigue, as well as the safety of the interventions, were assessed. Prior research has shown that treatment expectations can positively influence the effects of acupuncture. To control for these potential unspecific effects, we added expectations as a covariate [38].

## 2. Materials and Methods

### 2.1. Study Design

This study was conducted as a two-arm, single-center, randomized controlled trial, and the interventions were carried out between June and August 2019. Approval was obtained from the ethics committee of the University of Duisburg–Essen (project number: 18–8214–BO), and the study was registered at ClinicalTrials.gov (registration number: NCT03874598) prior to patient recruitment. The participants were recruited via e-mail obtained from newsletter lists of the Breast Unit, Evang. Kliniken Essen-Mitte, Essen, Germany. Eligibility was pre-screened in a telephone interview. The final inclusion of participants, fulfilling all inclusion and none of the exclusion criteria (see Table 1), was performed by the study physician. Before enrollment, participants received detailed written and verbal information about the study, and signed a consent form indicating their agreement to participate. The participants did not receive financial compensation for participation. The conduction of the study was in adherence to the tenets of the Declaration of Helsinki [39] and to good clinical practice, and the report was written in accordance with the CONSORT 2010 Statement [40] and the CONSORT harm extension [41].

### 2.2. Randomization

Patients were randomized to auricular acupuncture or psychoeducation after the first data collection by means of block randomization with randomly varying block lengths. The randomization list was generated by HC using a computerized randomization generator [42]. The list was password secured, and neither the study physicians nor the data collectors had access to it. Allocation concealment was achieved by central randomization by a study nurse so that investigators enrolling participants could not foresee allocation, and further treatment was carried out according to group allocation. Neither the patient nor the investigator had any influence on the randomization.

### 2.3. Interventions

Two licensed acupuncturists (Petra Voiss and Wiebke Kohl), with 20 and 5 years of experience, respectively, used the “very point” method to insert 0.20 × 15-mm TeWa PB-Type needles. The patients were asked on which side they would like to receive auricular acupuncture. After skin disinfection and inspection of the ear, the acupuncturist worked with a semi-standardized protocol and placed the needles in sensitive points—first on the postantitragal belt, second on the helix channel, and finally on the shen men. Additional points were used to address comorbid symptoms such as anxiety, hot flushes, and pain (Figure 1, Appendix A). The acupuncture needles were inserted 2–3 mm deep. Five to eight needles were used, and only one ear per patient was treated twice weekly for five weeks in a group setting of 6–10 participants. Patients were placed in reclining chairs in a group room at the hospital. The patients were asked to relax during the auricular acupuncture and keep quiet. The needles remained for at least 20 min. Comorbid conditions and adverse events were noted after each acupuncture session.

The control group received a single 90-min psychoeducation group session (8 patients) held by an experienced psychologist trained in psychoeducation/cognitive therapy for insomnia patients (Melanie D. Höxtermann). The program was based on the German S3 guidelines “Restless Sleep/Sleep Disorders” [43], and conveyed information concerning normal sleep, sleep problems, the use of sleep restriction in treating sleep problems, the use of stimulus control, psychohygiene, and cognitive behavioral elements that could be helpful in resolving sleep problems. Furthermore, all participants receiving psychoeducation received an advice booklet for dealing with sleep disorders [44].

### 2.4. Measures

All outcomes were assessed at weeks 0, 5, 17, and 29, except for IL-6—which was only assessed at weeks 0 and 5—and adverse events—which were continuously assessed during the study, and at weeks 5 and 17. Sociodemographic and clinical characteristics, as well as treatment expectations, were assessed at baseline.

### 2.5. Primary Outcome

#### Sleep Quality at Week 5

The primary outcome of this study was subjective sleep quality, measured by the Pittsburgh Sleep Quality Index (PSQI) [45] at the end of the intervention at week 5. The PSQI contains 19 self-assessment questions and 5 questions rated by the bed partner or roommate. The 5 questions rated by the bed partner or roommate are used as clinical information, and not included in quantitative analysis. The PSQI evaluates sleep quality in 7 domains over the past month (subjective sleep quality, sleep latency, sleep duration, habitual sleep efficiency, sleep disturbances, use of sleep medication, and daytime dysfunction). The total score is generated by the summation of the component scores, and can range from 0 to 21, with a higher score corresponding to a reduced sleep quality [45]. A cutoff score of 5 or 8 represents a “poor” sleeper in the general population or in cancer patients, respectively [46]. A change of 3 points or more on the PSQI is considered a minimal clinically important difference [47]. Brackhaus et al. showed the PSQI to have a high test–retest reliability and a good validity for patients with primary insomnia [48].

### 2.6. Secondary Outcomes

#### 2.6.1. Sleep Quality

For secondary outcomes, the PSQI was surveyed at week 17 and week 29.

#### 2.6.2. IL-6 Plasma Levels

IL-6 plasma levels were determined with the Human Th1/Th2 Cytokine Kit II (BD Biosciences, Heidelberg, Germany) at weeks 0 and 5. Capture beads were mixed, centrifuged for 5 min at 200× *g*, and the supernatant was aspirated and discarded. Bead pellets were vigorously resuspended in serum-enhancement buffer. For each sample—control and standard—50 µL of the capture bead suspension was added to a BD Falcon FACS tube (BD Biosciences, Heidelberg, Germany), followed by 50 µL of each standard, control, or ethylenediaminetetraacetic acid (EDTA) plasma sample and, subsequently, 50 µL of detection reagent. All tubes were incubated for 3 h at room temperature in the dark. After addition of 1 mL of wash buffer, samples were centrifuged for 5 min at 200× *g*. The supernatant was removed by aspiration, the bead pellets were resuspended in 300 µL of wash buffer, and the samples were acquired on a BD FACS Canto II with the FACS DIVA software 6.1.3 (both BD Biosciences, Heidelberg, Germany). Data were analyzed using the FCAP Array 3.0.19.2091 software (BD Biosciences, Heidelberg, Germany).

#### 2.6.3. Anxiety and Depressive Symptoms

The Hospital Anxiety and Depression Scale (HADS) is a self-report questionnaire consisting of 14 items that measure psychological wellbeing on the two scales depression HADS-D (7 items) and anxiety HADS-A (7 items), each with a sum score range of 0–21 [49]. Higher scores indicate greater psychological burden. The HADS has been shown to be a valid instrument for detecting clinically meaningful results as a psychological screening tool, and is sensitive to changes in response to therapies [50]. Recent research suggests an optimal cutoff of >9 for the HADS-A and of >7 for the HADS-D in cancer patients [51]. The HADS was assessed at weeks 0, 5, 17, and 29.

#### 2.6.4. Quality of Life and Fatigue

The Functional Assessment Of Cancer Therapy-Breast Cancer (FACT-B) [52] and Functional Assessment Of Chronic Illness Therapy-Fatigue (FACIT-F) [53] were secondary outcomes, and were assessed at weeks 0, 5, 17, and 29. The FACT-B uses the 27-item Functional Assessment of Cancer Therapy-General (FACT-G) questionnaire as a base, supplemented by 9 breast cancer items measured on a five-point Likert scale ranging from 0 (“not at all”) to 4 (“very much so”). Higher scores indicate a better quality of life [52,54,55]. The FACIT-F subscale consists of 13 fatigue items. Higher scores indicate less fatigue. The minimally important differences (MIDs) are 7–8 points for the FACT-B [56] and 3–4 points for the FACIT-F subscales [57,58]. Both have shown a good test–retest reliability and internal consistency [52,53].

#### 2.6.5. Stress

Perceived stress was assessed using the 10-item version of the Perceived Stress Scale (PSS) [59], rated for the past month on a 5-point rating scale (0 = “never”, 1 = “almost never”, 2 = “sometimes”, 3 = “fairly often”, 4 = “very often”). For the summed items (range from 0 to 40), a higher total score indicates greater stress, without a definite cutoff, because it is not a diagnostic instrument. The PSS has a good internal consistency and construct validity [60]. Perceived stress was assessed at weeks 0, 5, 17, and 29.

#### 2.6.6. Treatment Expectation at Week 0

Treatment expectation was measured via a 100-mm visual analogue scale (VAS) for expectations concerning treatment as “expecting the treatment to be not successful at all” and “expecting the treatment to be extremely successful”. All participants were asked before randomization to rate their expectations regarding both interventions separately. Only the expectation concerning the received treatment was used in the analysis.

### 2.7. Adverse Events

Adverse events were recorded by the acupuncturist or the psychologist, respectively, during each intervention using a standardized registration sheet. Furthermore, adverse events were assessed by enquiring self-reported adverse events of participants by week 5 and week 17 with open-ended questions. If the same adverse event was reported by the participant and the researcher, it was only accounted for once. Regardless of whether undesirable experiences during the course of the study were associated with the intervention or not, they were registered as adverse events. Serious adverse events were defined as: (1) death, (2) life-threatening situations, (3) hospitalization, (4) disability or permanent damage, (5) congenital anomaly/birth defect, or (6) the need for medical or surgical intervention [61,62]. All other adverse events were regarded as non-serious. If the licensed acupuncturist or psychologist suspected a causal relationship between the intervention and the adverse event, they were classified as intervention-related. Criteria for considering an adverse event to be intervention-related were: the specific adverse event had been previously reported in association with the intervention; it was clearly temporally related to the intervention; it occurred repeatedly in the intervention arm; and/or a causal relationship was physiologically plausible. The patient’s assessment was also taken into account in the evaluation. If patients dropped out due to adverse events, this was noted.

### 2.8. Sample Size Calculation

An a priori power analysis with an estimated effect size f of 0.315 [63] (Cohen’s d = 0.63), a 5% significance level, and 95% power for a mixed-model analysis of variance (ANOVA) with within- and between-subject interaction was performed. The analysis revealed that a total of 36 patients would be needed in order to detect an effect on the primary outcome. To compensate for a dropout rate of up to 30%, we planned to include a total of 52 patients in the study.

### 2.9. Statistical Analysis

All randomized patients were included in the analysis, regardless of whether they adhered to the allocated treatment and whether they had missing data. Missing data at all follow-up timepoints were multiply imputed. By using a multiple imputation technique, 20 additional datasets were generated using iterations based on multivariate regression models of sociodemographic and clinical parameters of baseline and outcome values. All analyses were based on the intention-to-treat sample and conducted with the Statistical Package for Social Sciences software (IBM SPSS Statistics for Windows, release 25.0; IBM Corporation, Armonk, NY, USA). A 2 (time: pre/post) × 2 (group: auricular acupuncture/psychoeducation) mixed-model ANOVA with treatment expectation as a covariate and sleep quality (total PSQI score) after treatment (week 5) as a dependent variable was conducted to test our primary hypothesis. Accordingly, secondary outcomes were evaluated in the same exploratory manner. To test moderation by the treatment expectation, the three-way interaction was also regarded. If a significant three-way interaction was found, a regression analysis with simple slope analyses was conducted. The non-parametric Fisher’s exact test was used in an exploratory fashion to test whether there was a significant difference between the good sleeper/bad sleeper groups (PSQI ≥ 8) at week 5, and on the clinically significant improvement of 3 points by the intervention group. A *p*-value < 0.05 was considered significant, and partial eta squared (*η*^2^_p_) was reported as an effect size estimator. According to Cohen, an *η*^2^_p_ of 0.01 is considered a small, of 0.06 a medium, and of 0.14 a large effect [64].

## 3. Results

### 3.1. Participant Characteristics

A total of 93 patients were assessed for eligibility (Figure 2); 52 patients fulfilled the inclusion criteria, and were randomized to either auricular acupuncture (*n* = 26) or psychoeducation (*n* = 26). For baseline demographics, see Table 2. All patients in the auricular acupuncture group attended at least 8 of the 10 appointments. Seventy-seven percent attended the one-time psychoeducation group. At week 5 and week 17, 94.2% of participants provided data, and 86.5% of participants at week 29.

### 3.2. Primary Outcome

A significant interaction effect with a medium effect size was found for condition x time on sleep quality (*p* = 0.031, *η*^2^_p_ = 0.094). Therefore, the auricular acupuncture in comparison with the psychoeducation had a significant effect on sleep quality over time (Figure 3). There was no statistically significant three-way interaction of expectation x time x group (*p* = 0.120, *η*^2^_p_  = 0.050), indicating no significant effect of treatment expectation. Therefore, we did not conduct further simple slope analyses.

At week 5, 19.2% of the control group and 61.5% of the intervention group had a PSQI score below 8 (*p* = 0.004). A clinically significant improvement of 3 points or more was observed in 38.5% of the control group and 65.4% of the intervention group (*p* = 0.095).

### 3.3. Secondary Outcomes

Auricular acupuncture significantly improved stress (*p* = 0.030; *η*^2^_p_  = 0.094), anxiety (*p* = 0.001; *η*^2^_p_ = 0.192), and fatigue (*p* = 0.006; *η*^2^_p_ = 0.148) at week 5 compared to psychoeducation. No significant group differences were found in IL-6, depressive symptoms, or quality of life at week 5. At week 17 and week 29, no significant group differences occurred on any outcome (Table 3, Figure 3).

### 3.4. Adverse Events

A total of 55 adverse events occurred in 20 patients during the study period. All adverse events occurred in the auricular acupuncture group. All in all, 3 patients (5.8%) reported 1, 7 patients (13.5%) reported 2, 5 patients (9.6%) reported 3, 2 patients (3.8%) reported 4, and 3 patients (5.8%) reported 5 adverse events. All adverse events were non-serious, and 39 (75%) of the events in 16 patients were judged to be intervention-related by the researchers (Table 4).

## 4. Discussion

This is the first trial to examine the effect of auricular acupuncture on subjective sleep quality in breast cancer survivors with insomnia. We found that the subjective sleep quality in the auricular acupuncture group compared to the psychoeducation group improved significantly over the study period (5 weeks), controlled for treatment expectation. Furthermore, auricular acupuncture improved stress, anxiety symptoms, and fatigue at week 5 compared to psychoeducation. No effect was found for depressive symptoms and quality of life at 5 weeks, or for any outcome at week 17 or week 29. Additionally, no effect on the inflammation parameter (IL-6 level) could be detected due to floor effects, as there was no elevation of IL-6 at baseline. No serious adverse events occurred during the study period.

In comparison to prior acupuncture trials concerning cancer survivors with insomnia, the mean global PSQI score declined to a similar degree [65], and even stronger than in a trial on electroacupuncture [66]. Therefore, our auricular acupuncture regime seems to have comparable effects to body acupuncture and electroacupuncture. While in our trial the semi-standardized auricular acupuncture was used, some prior trials used the standardized NADA (National Acupuncture Detoxification Association) protocol. In a randomized controlled trial, 59 insomnia disorder patients without cancer diagnosis were treated with either CBT-I or auricular acupuncture [67]. Significant between-group improvements were seen in favor of CBT-I. Both groups showed significant within-group post improvements, which were maintained after six months. Nevertheless, the authors concluded that auricular acupuncture cannot be considered an effective stand-alone treatment for insomnia disorders.

In our trial, three months after the intervention, the global PSQI score in the psychoeducation group further improved, while effects slightly diminished in the auricular acupuncture group, though still showing an improvement compared to baseline, with no significant differences between the groups (Table 3, Figure 3). The psychoeducation as well as the advice booklet for dealing with sleep disorders—which both include elements of CBT-I—might explain the improvement [23], though we did not query whether the booklet had been read. The first-line therapy for insomnia is CBT-I; unfortunately, certain factors complicate the access to insomnia treatment, including system barriers, clinician barriers, and patient barriers [15]. The system barrier of limited access to CBT-I and the patient barriers of time constraints and reluctance to engage in CBT-I were found to restrict the use of CBT-I, with only 5% effectively seeking treatment [68]. A broader range of evidence-based therapies to treat insomnia is needed in order to offer patient-centered supportive cancer care.

Therefore, for patients who reject CBT-I, auricular acupuncture might be an option [18], especially since fewer side effects are expected in comparison to pharmacological treatment.

All adverse events recorded were non-serious, and 39 (75%) of the events in 16 patients were classified by the investigators as being intervention-related. Non-serious adverse events reported at least twice included bruising, hunger pangs, flushing, heavy eyelids, fatigue, tenderness, and pain. To date, safety data on auricular acupuncture in cancer patients are scarce. In a recent trial, ear pain was the most common adverse event, and was reported by 18.9% of patients [34].

Additionally, the adherence to auricular acupuncture in our trial was high, with all 26 patients attending at least 8 of the 10 acupuncture group therapies, indicating a broad acceptance. Part of the effect of auricular acupuncture is attributed to a stimulation of the auricular branch of the vagus nerve, increasing parasympathetic activity by simultaneously decreasing sympathetic nervous system activity [69,70]; this may lead to improvements of stress-related symptoms. Alleviating comorbid conditions seems to influence patients’ perception of their ability to respond to acupuncture [24]. We took comorbid conditions such as hot flushes and pain into account using a semi-structured auricular acupuncture concept in order to choose additional individual acupuncture points. While many patients benefited from this regime, there were certain non-responders. The combination of acupuncture and sleep hygiene practice could enhance the treatment effect [24], and should be examined in future trials.

We were further interested in whether auricular acupuncture influences other health parameters, because breast cancer survivors who suffer from sleep disturbance often also experience fatigue and depression [71]. While we found a significant improvement in anxiety symptoms, fatigue, and stress at week 5, we did not find a significant improvement for depression or quality of life. Only 20% of the patients showed clinically relevant depressive symptoms in the HADS-D at baseline; therefore, it is difficult to detect improvements.

Despite several strengths, including sample size calculation and controlling for treatment expectation, there are a number of limitations to this study. First, the control group did not receive the recommended first-line therapy of CBT-I. Second, we did not use objective assessments for sleep quality, such as polysomnography. Third, we used a semi-standardized protocol for auricular acupuncture; while being a limitation, this was also a strength, since we were able to take individual comorbid conditions into account [24].

In future trials, auricular acupuncture should be compared with CBT-I using a non-inferiority design. To distinguish which patients benefit from which treatment, or to examine the effect of combining CBT-I with acupuncture, should be considered in future research.

Effective and easy-to-implement treatment options for cancer patients with insomnia need to be offered and integrated into the healthcare system.

## 5. Conclusions

Group auricular acupuncture might be an effective and safe intervention in treating sleep problems in breast cancer survivors in the short term, and may also reduce stress, anxiety, and fatigue. Long-term effects remain questionable.

## Figures and Tables

**Figure 1 cancers-13-04082-f001:**
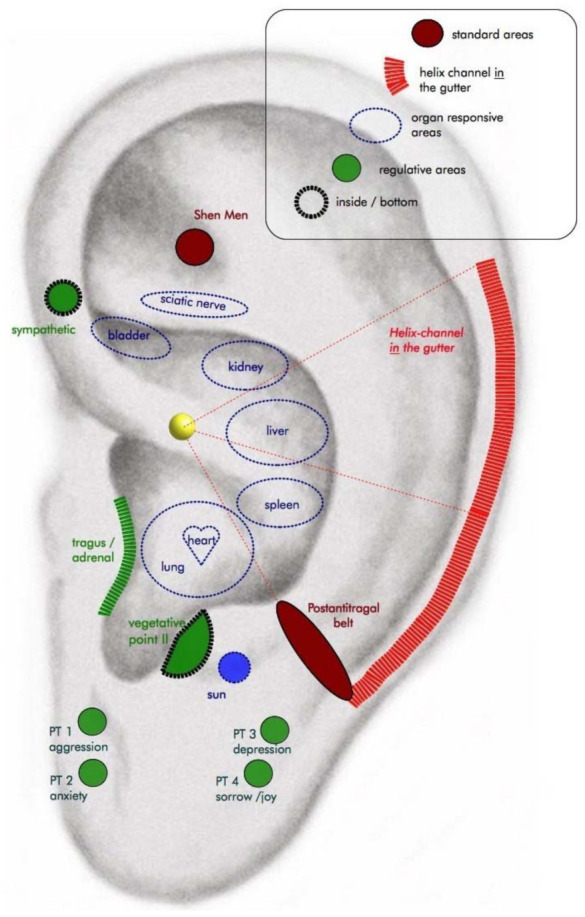
Insomnia auricular acupuncture protocol (adapted from Yase-Institut, Teaching institute for auricular acupuncture, Oldenburg, Germany).

**Figure 2 cancers-13-04082-f002:**
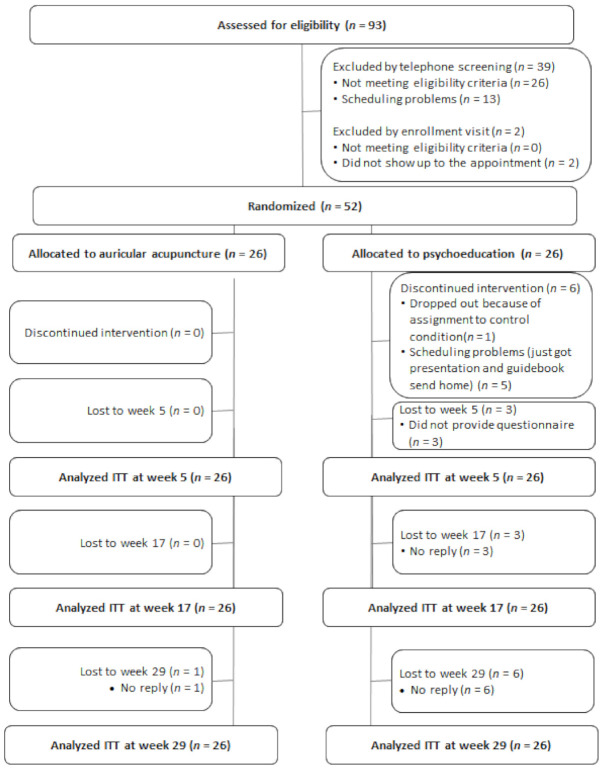
Study Flow. Abbreviation: ITT: intention-to-treat analysis.

**Figure 3 cancers-13-04082-f003:**
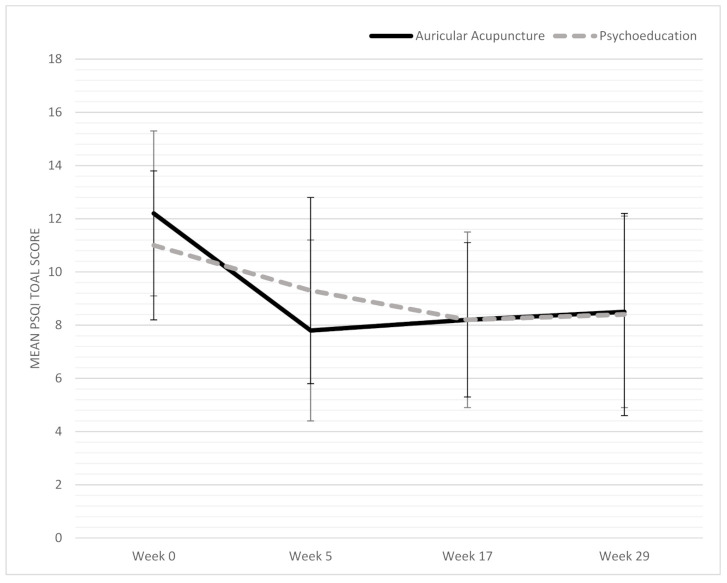
Change in sleep quality during the trial. Abbreviation: PSQI: Pittsburgh Sleep Quality Index. Lower scores indicate better sleep quality.

**Table 1 cancers-13-04082-t001:** Inclusion and exclusion criteria.

Inclusion Criteria	Exclusion Criteria
Histological diagnosed non-metastatic breast cancer (Classification of Malignant Tumors TNM stage I–III); Insomnia (diagnosed according to the Diagnostic and Statistical Manual of Mental Disorders DSM-5); Willingness to participate in the study (at least 8 out of 10 treatments if randomized to the auricular acupuncture group); Signed informed consent.	Ongoing or planned chemotherapy, radiation, follow-up treatment, or reconstructive plastic surgery during the study period; Severe physical or psychiatric comorbidity; Pregnancy; Participation in other clinical trials with behavioral, psychological, or complementary medical interventions during the study period; Regular use of barbiturates, antidepressants, or other sleep-inducing drugs; drug abuse; alcoholism.

**Table 2 cancers-13-04082-t002:** Baseline sociodemographic and clinical characteristics. Values are expressed as mean ± standard deviation, unless indicated otherwise.

Characteristics	Auricular Acupuncture (*n* = 26)	Psychoeducation (*n* = 26)	Total (*n*= 52)
Age (years)	56.58 ± 7.9	54.8 ± 8.3	55.73 ± 8.1
Weight (kg)	74.46 ± 14.9	72.7 ± 14.3	73.6 ± 14.5
Height (m)	1.7 ± 0.1	1.69 ± 0.1	1.7 ± 0.1
Time since diagnosis (months)	53.8 ± 35.2	57.8 ± 44.7	55.8 ± 39.9
Children	1.7 ± 1.0	1.3 ± 1.2	1.5 ± 1.1
Marital status *n* (%)			
Single/living alone	3 (11.5%)	3 (11.5%)	6 (11.5%)
Living with partner	1 (3.8%)	2 (7.7%)	3 (5.8%)
Married	19 (73.1%)	19 (73%)	38 (73.1%)
Divorced	1 (3.8%)	1 (3.8%)	2 (3.8%)
Widowed	2 (7.7%)	1 (3.8%)	3 (5.8%)
Education *n* (%)			
Secondary modern school (“Hauptschule”) qualification	2 (7.7%)	2 (7.7%)	4 (7.7%)
High school (“Realschule”) qualification	8 (30.8%)	6 (23.1%)	14 (26.9%)
A level (“Abitur”) (without subsequent studies/with uncompleted studies)	1 (3.8%)	6 (23.1%)	7 (13.5%)
University/college degree	15 (57.7%)	12 (46.2%)	27 (51.9%)
Employment *n* (%)			
Yes (full-time/part-time/training/retraining/sick leave currently under 6 months)	16 (61.5%)	18 (69.2%)	34 (65.4%)
No (housewife/unemployed /sick leave longer than 6 months)	2 (7.7%)	6 (23.1%)	8 (15.4%)
Retired	8 (30.8%)	2 (7.7%)	10 (19.2%)
TNM Stage * *n* (%)			
I	11 (42.3%)	13 (50%)	24 (46.2%)
II	10 (38.5%)	9 (34.6%)	19 (36.5%)
III	4 (15.4%)	4 (15.4%)	8 (15.4%)
Tumor biology *n* (%)			
Hormone-receptor-positive	22 (84.6%)	18 (69.2%)	40 (76.9%)
Triple-positive	0 (0%)	4 (15.4%)	4 (7.7%)
ER-positive and HER2-positive	1 (3.8%)	3 (11.5%)	4 (7.7%)
Triple-negative	3 (11.5%)	1 (3.8%)	4 (7.7%)
Prior chemotherapy *n* (%)	19 (73.1%)	14 (53.8%)	33 (63.5%)
Prior radiation therapy *n* (%)	23 (88.5%)	21 (80.8%)	44 (84.6%)
Prior operation *n* (%)	26 (100%)	24 (92.3%)	50 (96.2%)
Current antihormonal therapy *n* (%)	20 (76.9%)	17 (65.4%)	37 (71.2%)
Treatment expectancy **	7.81 ± 1.81	4.22 ± 2.96	6.02 ± 3.03

* For one patient in a non-metastatic situation in the auricular acupuncture group, only tumor biology is known. ** Treatment expectancy was rated for both potential treatments before randomization. Only the expectancy rating for the final allocated therapy is shown. Abbreviations: TNM Stage: Classification of Malignant Tumors; ER: Estrogen Receptor; HER2: Human epidermal growth factor receptor 2.

**Table 3 cancers-13-04082-t003:** Outcome measures.

Variable	Effect: Mean ± SD								Time × Group Interaction Effect					
	Auricular Acupuncture, *n* = 26				Psychoeducation, *n* = 26				Week 5		Week 17		Week 29	
	Week 0	Week 5	Week 17	Week 29	Week 0	Week 5	Week 17	Week 29	*p*	η^2^_p_	*p*	η^2^_p_	*p*	η^2^_p_
PSQI	12.2 ± 3.1	7.8 ± 3.4	8.2 ± 3.4	8.5 ± 3.6	11.0 ± 2.8	9.3 ± 3.5	8.2 ± 2.9	8.4 ± 3.8	0.031	0.094	0.141	0.045	0.133	0.046
PSS	20.2 ± 4.8	16.9 ± 4.0	18.0 ± 4.5	19.3 ± 4.6	18.0 ± 5.5	18.3 ± 5.8	18.4 ± 5.8	19.0 ± 5.3	0.030	0.094	0.079	0.063	0.902	0.000
FACT-B	92.2 ± 15.1	103.4 ± 12.7	97.5 ± 17.9	96.1 ± 17.9	96.3 ± 17.0	99.2 ± 17.2	96.9 ± 18.1	95.7 ± 19.9	0.158	0.041	0.494	0.010	0.700	0.003
FACIT-F	31.3 ± 10.5	40.6 ± 5.8	36.2 ± 11.7	35.3 ± 9.3	33.1 ± 10.8	34.3 ± 11.3	36.3 ± 10.3	34.3 ± 12.1	0.006	0.148	0.617	0.005	0.201	0.034
HADS-A	9.5 ± 4.2	6.0 ± 3.5	7.2 ± 4.0	7.5 ± 3.4	8.0 ± 4.3	7.8 ± 4.3	7.7 ± 4.0	8.4 ± 4.5	0.001	0.192	0.473	0.011	0.192	0.035
HADS-D	5.6 ± 3.5	3.4 ± 2.4	5.0 ± 3.5	5.0 ± 3.3	5.0 ± 4.2	5.3 ± 3.9	5.1 ± 3.9	5.6 ± 4.2	0.272	0.025	0.093	0.058	0.148	0.043
Interleukin 6	2.5 ± 2.2	2.2 ± 2.3	*	*	1.8 ± 2.5	1.4 ± 0.9	*	*	0.205	0.033	*	*	*	*

Abbreviations: PSQI: Pittsburgh Sleep Quality Index; PSS: Perceived Stress Scale; FACT-B: Functional Assessment of Cancer Therapy-Breast Cancer; FACIT-F: Functional Assessment of Chronic Illness Therapy-Fatigue; HADS-A: Hospital Anxiety and Depression Scale-Anxiety; HADS-D: Hospital Anxiety and Depression Scale-Depression; * Interleukin 6 was not assessed at the respective time points.

**Table 4 cancers-13-04082-t004:** Number of adverse events listed by treatment group.

Adverse Event	Auricular Acupuncture		Psychoeducation
	Non-Serious		Non-Serious
	Total	Intervention-Related	
Bruising	9	9	0
Pain	4	2	0
Pressure sensitivity	3	3	0
Hot flushes	6	6	0
Insatiable hunger/attacks of hunger/ravenous hunger	7	7	0
Restless legs syndrome	1	0	0
Flatulence/diarrhea	2	1	0
Cephalgia/tension headache	2	0	0
Fatigue	3	3	0
Sweats	1	0	0
Increased cramp tendency in the legs and thighs	2	0	0
Migraine attack	1	0	0
Swollen hand	1	0	0
Dental root inflammation	1	0	0
Lumbago	1	0	0
Abdominal discomfort	2	0	0
Xerostomia	1	0	0
Dysgeusia	1	0	0
Dull feeling in the head	1	1	0
Unrest	1	1	0
Itching on the ear	1	1	0
Heavy eyelids	4	4	0

## Data Availability

Data are available on request from the authors.

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
