# Peer review of "Efficacy and Safety of Auricular Acupuncture for the Treatment of Insomnia in Breast Cancer Survivors: A Randomized Controlled Trial"

_cancers, 2021, doi:10.3390/cancers13164082_

Round 1

Reviewer 1 Report

Congratulation to all research team members in this project. This is a great article with big contribution in oncology care. I am sure many breast cancer patients will get benefits from this scientific paper. 

Author Response

Dear Reviewer,

thank you for the appreciative feedback!

Reviewer 2 Report

With respect to the manuscript “Efficacy and safety of auricular acupuncture for the treatment of insomnia in breast cancer survivors: a randomized controlled trial”, I think that the subject is very interesting and of importance for health professionals of the oncology area.

However, I think that clarification about various aspects of the trial is needed:

1. Revision of the English of the text to make it clearer.

2. The Introduction should include a more extended revision on auriculotherapy and its application in oncology.

3. Authors should provide more information about the methodological aspects of the study:

    1. The patients were naïve about acupuncture?
    2. A clearer definition of the auricular points is needed.
    3. Which ear was acupunctured? The dominant one?
    4. The treatment lasted 20 or 30 minutes, based on which criteria?

4. I think it would be very important to refer in the manuscript which are the additional points used. Maybe the authors could present in a table the protocol and results of each patient who received the auriculotherapy.

5. Table 3 should be modified in order to be more comprehensive.

6. Authors should explain what were the criteria to consider an adverse event as intervention related or not.

7. The limitations of the study, namely the lack of standardization of an auriculotherapy protocol, should be referred in the Abstract.

8. Taking into account the limitations of the study, namely the lack of standardization of an auriculotherapy protocol, conclusions should be more cautious.

Reviewer 3 Report

Thank you for all your hard work on this research. It is a very interesting study. However, the study did not include patients or acupuncturist blinding methods and might be affected by the placebo effect, the study sample size is also not considered large. There are a few issues and questions that need to be addressed:

Q1: Does only one cognitive behavioral therapy is sufficient in order to generate the therapeutic effect?

Q2: Line 68: The mark - after the word time-, is this a typing mistake?

Q3: Line 80: The study was conducted between June and August 2019, is the time correct? If it is correct how does only 2 months of study allow for the long follow up.

Q4: line 104: what does “with 20 respectively 5 years of experience” mean? 20 or 5 years, or with 20 and 5 years respectively?

Q5: In the introduction some of the citation are in the middle of the sentence, please check if it appropriate to move the citation to the end of the sentence.

Q6: For the acupuncture method, can you explain the term “a semi-standardized program” (line 105).

Q7: “First needle in the region polster, the second needle in the autonomic channel”, please provide the acupuncture point full name and number. Please also provide a picture of the acupoint in a figure in manuscript or as a supplementary. Please also describe the acupuncture stimulation method and if the “De Chi” was achieved following the “STRICTA checklist”. Please name the rest of the points used as well.

Q8: Did the patients in the treatment group also received the 90-minute psychoeducation group session?

Q9: Does a multiple imputation technique of 20 additional data sets contribute to study bias?  Why 20 data sets if only 7 data sets were missing? And to which follow up time was the multiple imputation implemented on?

Q10: Why treatment expectation was used as a covariate? And can data on treatment expectations be presented in a Table?

Q11: Is it possible to add P-value for Table 2? And to add P-value of week 0 on Table 3 (between the 2 groups groups)?

Q12: Please add a figure legend for Table 3,4.

Q13: Table 4: What is the meaning of “Flavor disorders”?

Q14: Is it possible to add data on cancer stage to Table 2?

Round 2

Reviewer 2 Report

With respect to the manuscript “Efficacy and safety of auricular acupuncture for the treatment of insomnia in breast cancer survivors: a randomized controlled trial”, I think that the subject is very interesting and of importance for health professionals of the oncology area.

In the present version of the manuscript, authors incorporate previous recommendations. In my opinion, with the modifications made in the work the manuscript has now a better quality.  I think that in the present form, this work can bring a contribution to the efficiency and safety of auriculotherapy for symptoms suffered by oncology patients.